# Nuclear Nestin deficiency drives tumor senescence via lamin A/C-dependent nuclear deformation

Yanan Zhang[1,2], Jiancheng Wang[1,2], Weijun Huang[2], Jianye Cai[2,3], Junhui Ba[4], Yi Wang[1,2], Qiong Ke[2], Yinong Huang[1,2], Xin Liu[1,2], Yuan Qiu[1,2], Qiying Lu[1,2], Xin Sui[1,2], Yue Shi[1,2], Tao Wang[1,2], Huiyong Shen[5], Yuanjun Guan[6], Ying Zhou[7], Yuan Chen[8], Maosheng Wang[9] & Andy Peng Xiang[1,2,10]

Emerging evidence has revealed that Nestin not only serves as a biomarker for multipotent stem cells, but also regulates cell proliferation and invasion in various tumors. However, the mechanistic contributions of Nestin to cancer pathogenesis are still unknown. In the present study, previously thought to reside exclusively in the cytoplasm, Nestin can also be found in the nucleus and participate in protecting tumor cells against cellular senescence. Specifically, we reveal that Nestin has a nuclear localization signal (aa318–aa347) at the downstream of rod domain. We then find nuclear Nestin could interact with lamin A/C. Mechanistic investigations demonstrate that Nestin depletion results in the activation of cyclin-dependent kinase 5 (Cdk5), which causes the phosphorylation of lamin A/C (mainly at S392 site) and its subsequent translocation to the cytoplasm for degradation. The findings establish a role for nuclear Nestin in tumor senescence, which involves its nucleus-localized form and interaction with lamin A/C.

[1] Program of Stem Cells and Regenerative Medicine, Affiliated Guangzhou Women and Children's Hospital, Zhongshan School of Medicine, Sun Yat-sen University, Guangzhou, 510623, China. [2] Center for Stem Cell Biology and Tissue Engineering, Key Laboratory for Stem Cells and Tissue Engineering, Ministry of Education, Sun Yat-Sen University, Guangzhou, 510080, China. [3] Department of Hepatic Surgery and Liver transplantation Center of the Third Affiliated Hospital, Organ Transplantation Institute, Sun Yat-sen University, Guangzhou, 510630, China. [4] Department of Medical Intensive Unit, The Third Affiliated Hospital of Sun Yat-sen University, No. 600 Tianhe Road, Guangzhou, 510630, China. [5] Department of Orthopedics, Sun Yat-sen Memorial Hospital, Sun Yat-sen University, Guangzhou, 510120, China. [6] Core Facility Center, Zhongshan Medical School, Sun Yat-Sen University, Guangzhou, 510080, China. [7] Department of Nephrology, The First Affiliated Hospital of Wenzhou Medical University, Wenzhou, 325000, China. [8] Center for Neurobiology, School of Medicine, Sun Yat-Sen University, Guangzhou, 510080, China. [9] The Cardiovascular Center, Gaozhou People's Hospital, Maoming, 525200, China. [10] Guangzhou Regenerative Medicine and Health Guangdong Laboratory, Guangzhou, 510080, China. These authors contributed equally: Yanan Zhang, Jiancheng Wang, Weijun Huang.  Correspondence and requests for materials should be addressed to M.W. (email: mmwmsmd@126.com) or to A.P.X. (email: xiangp@mail.sysu.edu.cn)

Nestin, a type VI intermediate filament (IF) protein, is originally identified as a marker for neural stem cells in early development[1,2]. In adult tissues, most Nestin-positive cells are found in areas of stem/progenitor populations, like hair follicle[3-5], skeletal muscle satellite cells[6], testis[7], kidney[8], and bone marrow[9], where they might be engaged in active proliferation, tissue regeneration, and wound healing[10]. In addition, Nestin can play a role in pathogenesis and it is expressed in several types of malignancies, including glioma[11], melanoma[12], gastrointestinal tumors[13], prostate cancer[14], and so on. Furthermore, higher levels of Nestin expression seems to correlate with greater malignancy and poorer prognosis[11-15].

Although several studies reveal the involvement of Nestin in tumor cell migration, invasion, and metastasis, the roles and molecular mechanisms of Nestin expression in cancers remain elusive. Hyder et al.[16] showed Nestin regulates prostate cancer cell invasion by influencing spatial FAK activity, integrin's cell membrane localization and dynamics, and extracellular matrix proteolysis. Moreover, Li et al.[17] found that Nestin cooperates with Hedgehog (Hh) signaling to drive medulloblastomas growth through blocking the Hh pathway transcription factor-Gli3 phosphorylation and its subsequent proteolytic processing. Recently, our study demonstrated that Nestin can also regulate proliferation and invasion of gastrointestinal stromal tumor cells by recruiting dynamin-related protein1 to alter mitochondrial dynamics[13], indicating Nestin may not only participate in processing signal transduction, motility, and cellular stress but also play a role in regulating spatial localization of cell organelles.

In the past, Nestin was thought to be a cytoplasmic protein, but recent studies revealed that Nestin localized to the nucleus as well. For example, Nestin has been observed in the nucleus of glioblastoma and neuroblastoma cells[18,19]. Our previous results also revealed Nestin expression in the nuclei of lung carcinoma cells[20]. Recently, the proteomic analysis of Nestin-knockdown glioblastoma cells demonstrated that suppression of Nestin dramatically decreases expression of prelamin-A/C[21], which are bona fide nuclear proteins responsible for the meshwork covering the inner surface of the nuclear envelope[22]. Accordingly, it will be interesting to clarify whether Nestin is a nucleocytoplasmic shuttling protein, how Nestin participates in the regulation of lamina stability and what is functional significance of nuclear-localized Nestin? In the present study, using non-small-cell lung carcinoma (NSCLC) model cell lines, we investigate the nuclear localization and functional roles of Nestin and reveal Nestin can import into the nucleus through a classical nuclear localization signal (NLS). We further show that Nestin stabilizes lamin A/C for maintaining nuclear integrity and protecting tumor cells from senescence.

## Results

**Nestin deficiency drives nuclear deformation and tumor senescence.** Nestin is an IF protein whose expression is upregulated in numerous cancers, and is correlated with aggressive behavior and poor prognosis[12,14,23]. To identify the mechanistic contributions of Nestin to tumor pathogenesis, we used short hairpin RNAs (shRNAs) to deplete Nestin in the lung cancer cell lines, A549 and H1299. Two independent Nestin shRNAs showed consistent and significant effects (Supplementary Fig. 1a, b). Surprisingly, Nestin-knockdown cells frequently exhibited nuclear malformations (Supplementary Fig. 1c), which is an important biomarker of cellular senescence[24]. To further image nuclear deformation, we used tumor cells genetically labeled with green fluorescent protein (GFP) in the nucleus and red fluorescent protein (RFP) in the cytoplasm[25-27]. Consistently, Nestin-knockdown cells exhibited obvious nuclear malformations

(Fig. 1a). Subsequently, we calculated three typical examples of nuclear shape alterations, specifically budded nuclei[28-30], and found that Nestin-knockdown cells had a significantly increased fraction of abnormally shaped nuclei (Fig. 1b and Supplementary Fig. 1d). To more quantitatively assess the degree of irregular nuclear shape, we computed the nuclear circularity ($4\pi \times$ area perimeter$^{-2}$) of each cell group. For a circular shape, the nuclear circularity had a value of 1; less nuclear roundness was associated with smaller values, and an altered nuclear envelope was defined as circularity $\leq 0.65$[28,30-32] (Fig. 1c). Indeed, Nestin knockdown significantly decreased the mean circularity and increased the percentage of abnormally shaped nuclei (circularity $\leq 0.65$) compared with control cells (Fig. 1c and Supplementary Fig. 1e). Accompanied with an alteration of the nuclear architecture, the number of Nestin-knockdown cells was dramatically reduced when compared to control cells cultured under the same conditions (Fig. 1d). The lower number of Nestin-knockdown cells was not a consequence of increased apoptosis but cell cycle arrest (Supplementary Fig. 1f, g), which is another important indication of cellular senescence[24]. Meanwhile, we observed that knockdown of Nestin in A549 and H1299 cells increased the protein levels of p16 and p21, which are critical activators of the senescence program[33] (Supplementary Fig. 1h). Consistently, Nestin knockdown could decrease the phosphorylation levels of pRb and upregulate p53 (except for H1299, the p53-null cell line), which are broadly considered to be involved in senescence growth arrest[33] (Supplementary Fig. 1i). To investigate whether Nestin knockdown could trigger cellular senescence, we further measured senescence-associated acidic β-galactosidase (SA-β-gal) staining and found that Nestin knockdown increased the frequency of SA-β-gal-positive cells (Fig. 1e, f). Consistent with this observation, Nestin-knockdown cells exhibited upregulation of pro-inflammatory cytokines (interleukin (IL)-6 and IL-8), which reflects activation of the senescence-associated secretory phenotype[33] (Fig. 1g). Furthermore, immunoblotting and immunostaining of H3K9me3 revealed a global reduction of heterochromatin structure in Nestin-knockdown cells (Fig. 1h, i). Together, these findings indicate that Nestin-knockdown cells might have shown treatment-induced senescent phenotypes.

**Nestin stabilizes lamin A/C to protect tumor cells from senescence.** Since tumor cell senescence caused by Nestin depletion was associated with alteration of nuclear shape, we speculated whether Nestin plays a role in maintaining the nuclear architecture. It is well known that lamins positioned beneath the inner nuclear membrane (INM) are keys to maintaining the structural integrity of the nuclear shape[34]. To address whether the misshapen nuclei of Nestin-knockdown cells reflected an alteration of lamins, we evaluated the expression of lamins by quantitative PCR and immunoblotting in the above-mentioned lung cancer cell lines. We found that Nestin knockdown did not affect the mRNA levels of any lamin tested (Fig. 2a). Meanwhile, Nestin knockdown significantly decreased the protein levels of endogenous lamin A/C but did not affect the protein levels of lamin B (Fig. 2b). In addition, our immunostaining analysis revealed that lamin A/C was loosely dispersed beneath the INM of Nestin-knockdown cells (Fig. 2c), indicating that Nestin might affect the stability of lamin A/C. Consistently, overexpression of lamin A/C restored the nuclear shape and attenuated senescence in Nestin-knockdown cells, supporting the involvement of lamin A/C in the observed phenotype (Fig. 2d–i). These results suggest that Nestin depletion triggers a progressive disorganization and degradation of lamin A/C.

To further confirm the overall influence of Nestin on cellular senescence and lamin A/C expression, we knocked out *NES* in the

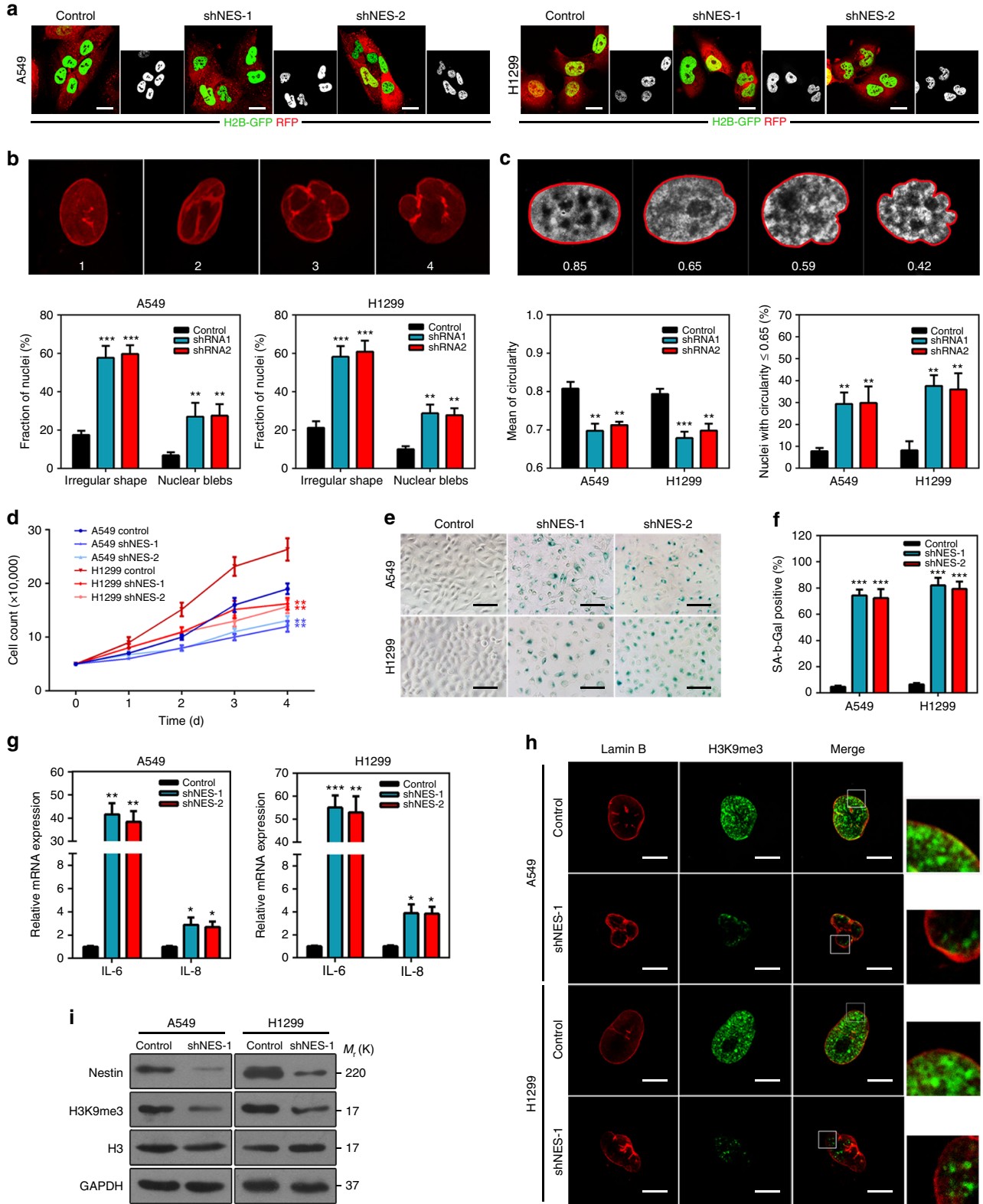

H1299 cell line with CRISPR/Cas9 system, which has been reported to efficiently disrupt genes in various organisms[35]. Only one of the three designed guide RNAs (gRNAs) was able to mediate the efficient introduction of indel mutations at its target site, as assessed using a previously described T7 endonuclease 1 (T7E1) assay[36] (Supplementary Fig. 2a, b). The functional gRNA was selected for further analysis (Supplementary Fig. 2c). Consistent with the above-described results, the fraction of abnormally shaped nuclei was significantly increased in H1299 cells transfected with Cas9/gRNA, as assessed 4 days post

**Fig. 1** Nestin knockdown induces cellular senescence accompanied by nuclear deformation. **a** The nuclear shapes of control and Nestin-knockdown cells observed with dual-color labeled. Scale bars, 20 μm. **b** Analysis of nuclear morphology. The upper panels showed examples of normal nuclei (1) and the abnormal crumpled nuclei (2), nuclear envelope lobulations (3), and nuclear blebbing (4) observed in Nestin-knockdown lung cancer cells. The lower panels showed that the fractions of abnormally shaped nuclei and nuclear blebs were significantly increased in Nestin-deficient cells. **c** Nuclear circularity was analyzed using the Cellprofiler software. The upper panels showed examples of circularity values obtained for nuclei, with a value ≤ 0.65 defined as representing an abnormal shape. The lower panels showed mean circularities in control and Nestin-knockdown lung cancer cell lines (left) and percentage of cells with deformed nuclei (right, circularity ≤ 0.65). **d** Proliferation of A549 cells and H1299 cells, as detected by a cell-counting assay. **e** Senescence-associated β-galactosidase staining was used to identify the frequency of senescent cells in vitro (left panel, ×400). Scale bars, 100 μm. **f** Quantification of the SA-β-gal-positive cells in **e**. **g** qPCR was used to assess the expressions of the SASP factor genes, IL-6 and IL-8, in Nestin-knockdown and control cells. **h** Immunofluorescence staining of H3K9me3 (green) and lamin B (red) in control and Nestin-knockdown cells. Scale bars, 10 μm. **i** Representative immunoblotting showing the levels of H3K9me3 in total cell lysates from Nestin-knockdown and control cells. GAPDH and H3 were used as cytoplasmic and nuclear marker proteins to confirm the quality of the fractions. The quantified results were presented as mean ± SEM of three independent experiments, as assessed using two-way ANOVA (**d**) and unpaired *t*-test (**b**, **c**, **f**, **g**). *$P < 0.05$, **$P < 0.01$, ***$P < 0.001$

transfection (Supplementary Fig. 2d, e). Moreover, *NES* knockout considerably increased the protein levels of p16 and p21 and enhanced SA-β-gal activity, indicating that cellular senescence was elevated. Conversely, overexpression of lamin A/C rescued the senescence and lamin A/C status of Nestin-depleted cells (Supplementary Fig. 2f–h). Together, these results show that Nestin appears to protect tumor cells from senescence by stabilizing lamin A/C.

**Sequences downstream of the rod domain are required for the nuclear import of Nestin**. As Nestin is considered as a predominantly cytoplasmic protein, whereas lamin A/C is nuclear, we questioned how Nestin could modulate the stability of lamin A/C. To determine the underlying mechanism, we first observed the subcellular localization of Nestin in vitro, and found that Nestin was present in the nuclei of the tested lung cancer cell lines (Fig. 3a–c). Next, we constructed mouse xenograft models to examine the subcellular localization of Nestin in vivo. Consistent with in vitro results, we detected Nestin in the nuclei of cells xenografted to nude mice (Supplementary Fig. 3a, b). Immunohistochemical analysis of specimens from NSCLC patients also supported the presence of Nestin in the nuclei of lung cancer cells (Supplementary Fig. 3c). Generally, NLSs are necessary and sufficient to target a protein over 50 kDa to the nucleus. Classical NLSs fall into two categories: a simple sequence of 3–5 basic amino-acid residues; and a bipartite signal comprising the simple sequence plus a basic dipeptide 10–25 residues upstream of the simple sequence[37]. Interestingly, our bioinformatic analysis identified three putative NLSs (NLS1: aa70–aa100, NLS2: aa318–aa347, and NLS3: aa734–aa767) within the Nestin sequence. Accordingly, we constructed six expression vectors, including one, two, or all three of the potential NLSs fused to GFP and investigated their subcellular localizations by confocal fluorescence microscopy (Fig. 3d). We found that GFP fused to the F6 fragment, which contained NLS2 alone, could localize exclusively to the nucleus. GFP fused to the F1 and F4 fragments also yielded nuclear signals, though at lower levels. In addition, GFP fused to F2 fragment was deposited at the periphery of outer nuclear envelope. However, GFP fused to fragment contains NLS1 or NLS3 alone localized to the cytoplasm (Fig. 3e and Supplementary Fig. 3d). These results, which suggest that Nestin contains an intrinsic NLS (NLS2), were further verified by immunoblotting (Fig. 3f). We also transfected A549 and H1299 cells with a mutant form of GFP-F6, GFP-M6, which harbored mutations at three vital basic amino-acid sites (K330A, R342A, and R343A) of the NLS. As expected, this fusion showed a decreased ability to translocate into the nucleus (Fig. 3g, h). Moreover, our analysis of several other cancer cell lines showed that Nestin universally underwent nuclear translocation in cancer cells (Supplementary Fig. 3e, f). Together, these findings

demonstrate that Nestin contains a NLS (aa318–aa347) at the downstream of Nestin rod domain (aa9–aa313) and can localize to the nucleus for its further functions.

**The rod domain of Nestin mediates its interaction with lamin A/C**. The nuclear import of Nestin suggested that it could directly regulate lamin A/C-dependent nuclear deformation and tumor senescence. To test this possibility, we first overexpressed and immunoprecipitated Flag-Nestin in lung cancer cells. As shown in Fig. 4a, lamin A/C co-precipitated with Nestin. A direct interaction between lamin A/C and Nestin was also observed by reciprocal immunoprecipitation with lamin A/C antibody (Fig. 4b). Furthermore, super-resolution structured illumination microscopy (SIM) revealed that Nestin and lamin A/C co-localized beneath the INM of lung cancer cells (Fig. 4c and Supplementary Movie 1 and 2). Together, these data demonstrate that there is a direct interaction between nuclear lamin A/C and Nestin. To identify the exact region within Nestin that is responsible for mediating its physical binding with lamin A/C, we generated various constructs based on the structural composition of Nestin (Fig. 4d). The domain-mapping studies revealed that the C terminus of the Nestin rod domain (aa182–aa313) retained the ability to bind lamin A/C (Fig. 4e), which is consistent with previous reports that the rod domain region is critical for the interactions of IF proteins with their binding partners. When we evaluated the effects of lamin A/C downregulation on Nestin expression, we found that lamin A/C depletion had no effect on the mRNA levels of Nestin (Fig. 4f), but significantly decreased its protein levels (Fig. 4g). Together, these results indicate that Nestin and lamin A/C appear to interact with each other beneath the INM.

In addition to its impact on Nestin protein expression, shRNA-mediated lamin A/C depletion also caused nuclear shape defects (Supplementary Fig. 4a,b), as previously reported[38]. Moreover, these cells recapitulated major phenotypes of premature senescence, including a flat and enlarged morphology, increased SA-β-gal positivity, and downregulation of the constitutive heterochromatin marker, H3K9me3 (Supplementary Fig. 4c–e).

**Nestin protects lamin A/C from proteasomal degradation**. To further investigate the relationship between Nestin and lamin A/C, we then detected the expression of lamin A/C in control and Nestin-knockdown cells treated with the translational inhibitor cycloheximide (CHX). Immunoblotting revealed that lamin A/C was more highly degraded in Nestin-knockdown A549 cells compared to controls (Fig. 5a), and our statistical analysis showed that lamin A/C had a shorter half-life in Nestin-knockdown A549 cells relative to controls (Fig. 5b). These results demonstrate that Nestin affects lamin A/C stability per se. Next, we sought to

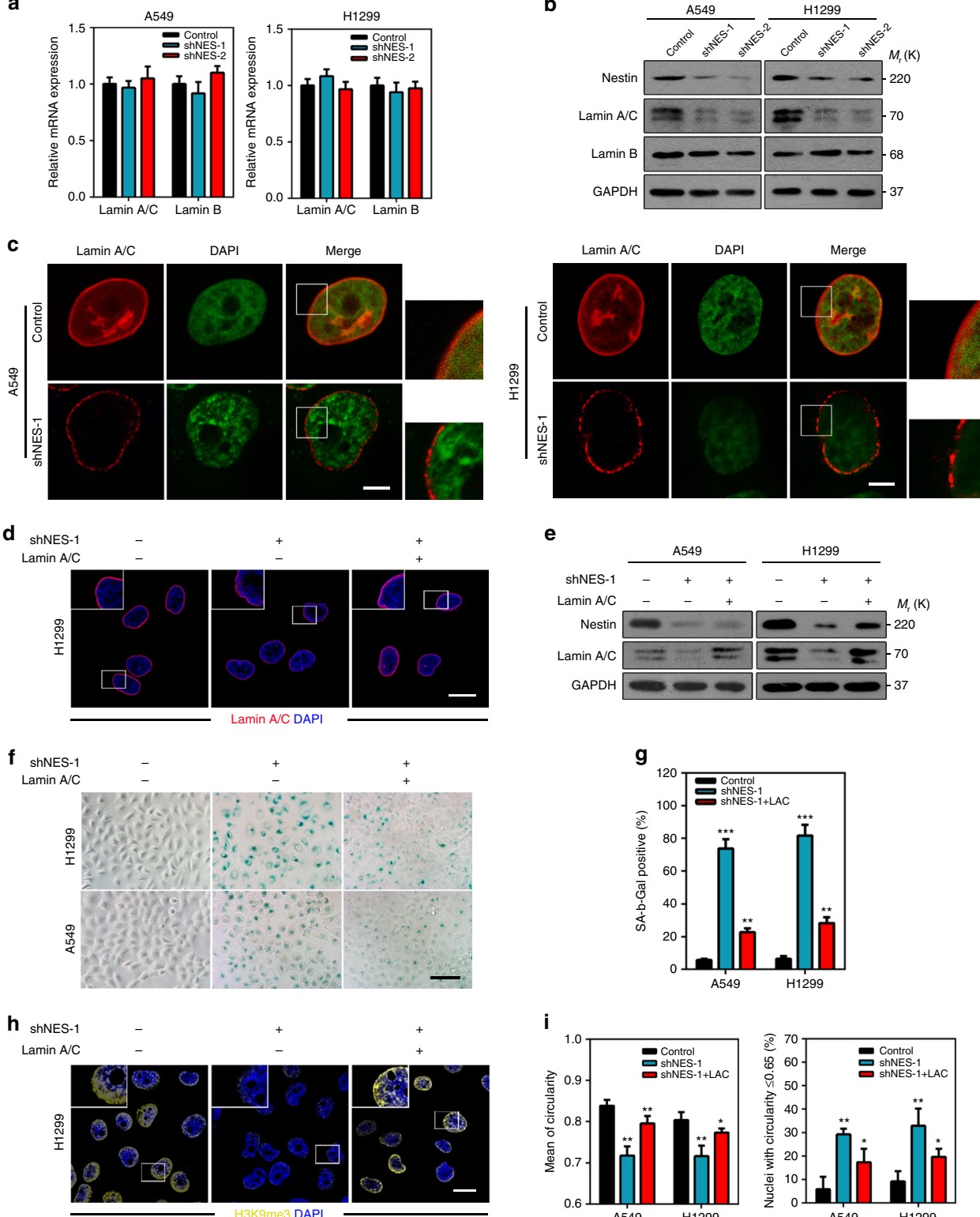

**Fig. 2** Nestin expression is required for the stability of lamin A/C and evasion of senescence in cancer cells. **a** qPCR analysis of the expression of lamins in control and Nestin-knockdown cells. **b** Immunoblotting analysis of lamins expression in control and Nestin-knockdown cells. **c** Immunofluorescence staining of lamin A/C in control and Nestin-knockdown cells. Scale bars, 5 μm. **d** Immunofluorescence staining of lamin A/C in control, Nestin-knockout, and Nestin-knockout/lamin A/C-rescue H1299 cells. Scale bars, 20 μm. **e** Immunoblotting analysis of lamin A/C in control, Nestin-knockout, and Nestin-knockout/lamin A/C-rescue cells. **f** SA-β-gal staining was used to identify the frequency of senescent cells in vitro (left panel, ×400). Scale bars, 100 μm. **g** Quantification of the SA-β-gal-positive cells in **f**. **h** Immunofluorescence staining of H3K9me3 in control, Nestin-knockout, and Nestin-knockout/lamin A/C-rescue H1299 cells. Scale bars, 20 μm. **i** Nuclear circularity was analyzed with the Cellprofiler software. Shown were the quantifications of mean circularity and the percentage of cells with deformed nuclei in cells subjected to Nestin knockdown and lamin A/C overexpression. For all quantifications, the data represented the mean ± SEM of three independent experiments, as assessed using unpaired $t$-test. $*P < 0.05$, $**P < 0.01$, $***P < 0.001$

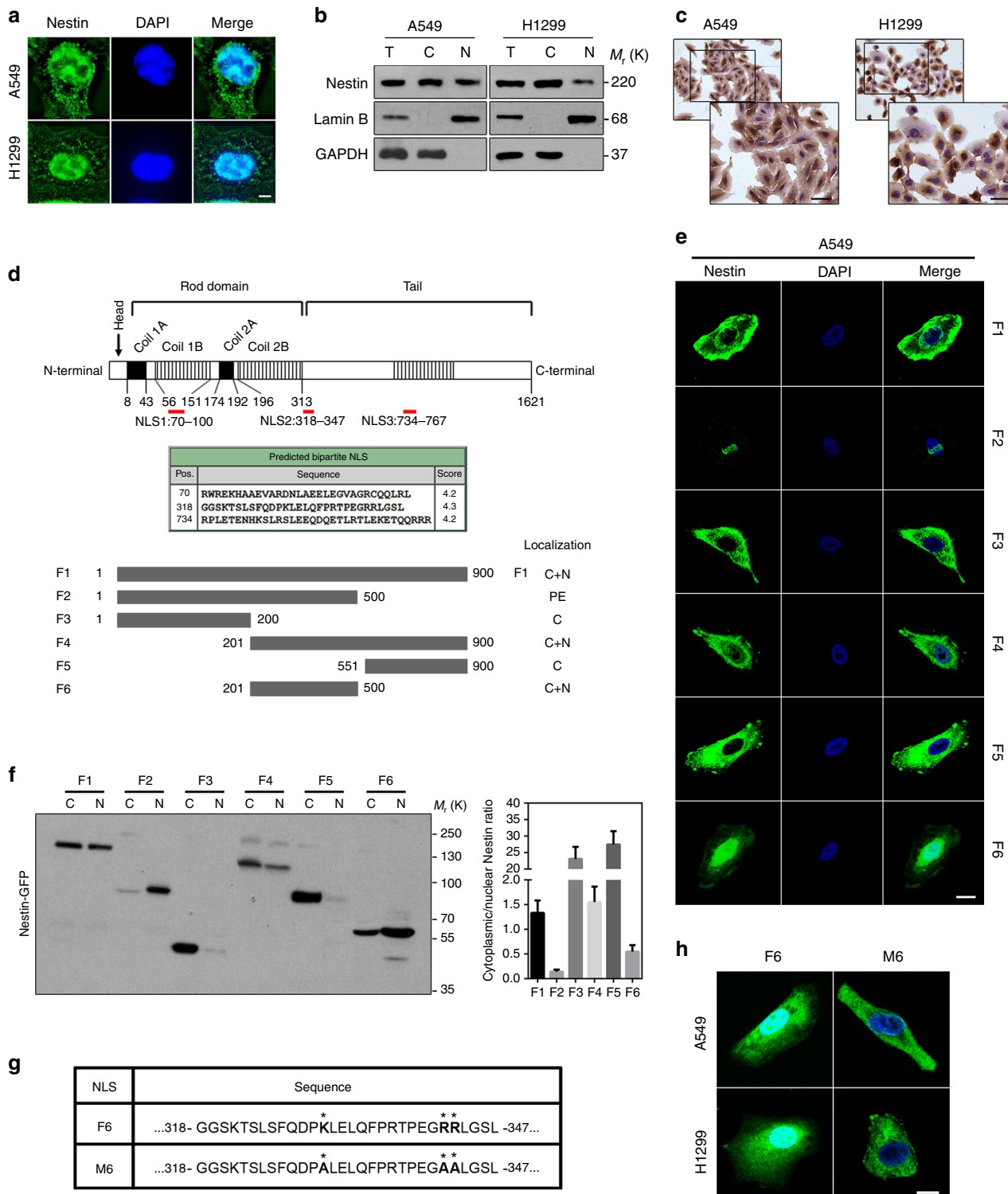

**Fig. 3** Nestin contains a nuclear localization signal (NLS). **a** Confocal images of Nestin immunostaining (green) in A549 and H1299 cells. Scale bars, 5 μm. **b** Immunoblotting for subcellular fractions of Nestin in A549 and H1299 cells (T total cell lysates, C cytoplasmic cell lysates, N nuclear cell lysates). **c** Immunohistochemical staining of Nestin in A549 and H1299 cells. Scale bars, 30 μm. **d** Upper: three putative NLSs identified by bioinformatic analysis. Lower: schematics of the Nestin constructs used for the localization study. Each fusion protein was expressed with a GFP tag (left), and its localization was assessed (right; C, predominantly cytoplasmic; C + N, in both compartments; PE, at the periphery of outer nuclear envelope). **e** Confocal images of GFP-Nestin fusion protein in A549 cells. Nuclei were marked by DAPI (blue). Scale bars, 20 μm. **f** Immunoblotting analysis of GFP-Nestin distribution in A549 cells (left; C cytoplasmic cell lysates, N nuclear cell lysates) and the corresponding cytoplasm/nucleus intensity ratio (CN ratio) of GFP-tagged proteins (right). **g**, **h** Schematics of the GFP-tagged Nestin constructs showing the sequences (**g**) and positions (**h**) of a potential NLS and the targeted aa substitutions (asterisk) used to disable it. Scale bars, 10 μm

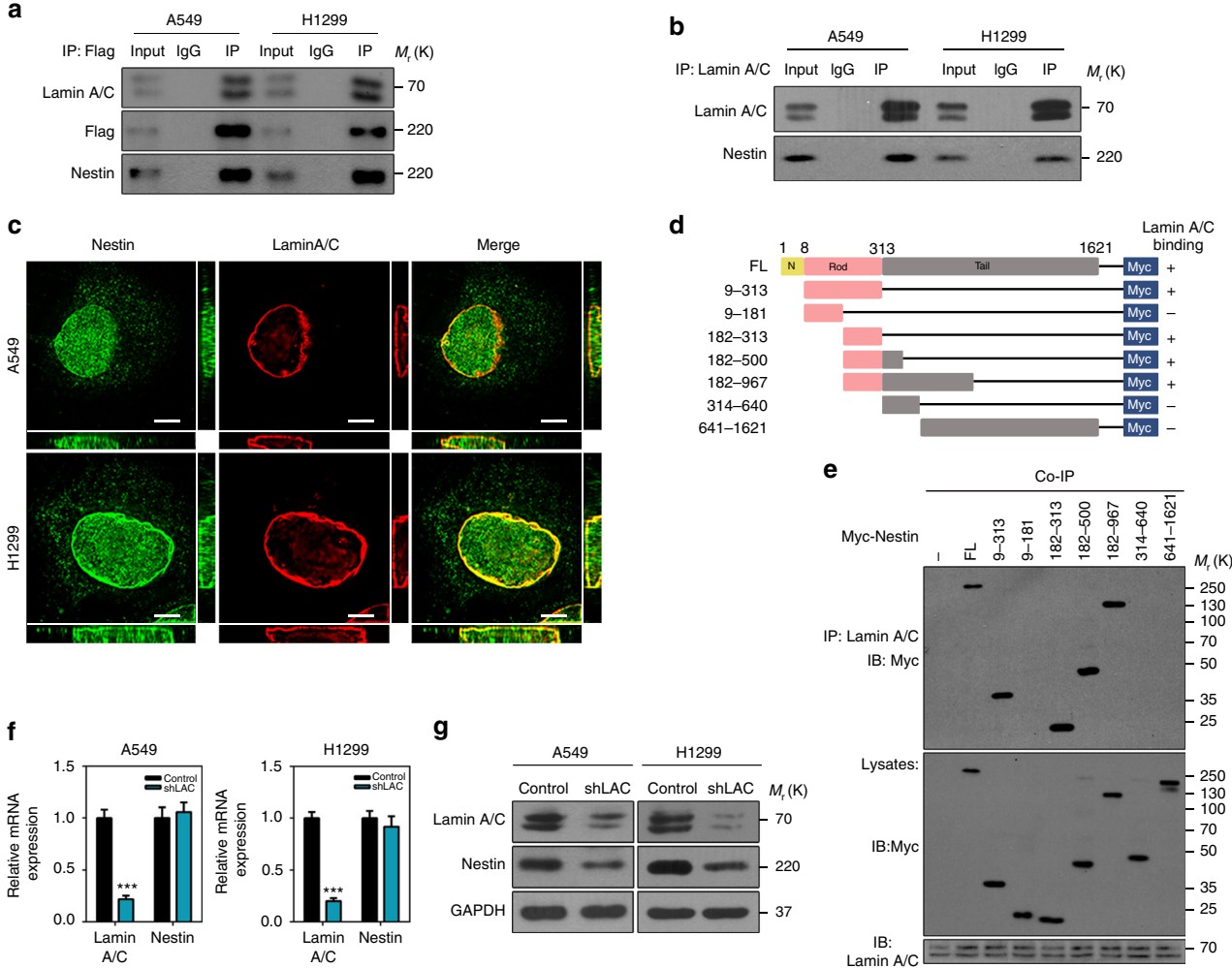

**Fig. 4** Nestin interacts with lamin A/C. **a** Coimmunoprecipitation of Flag-Nestin and lamin A/C from A549 and H1299 cell lysates with anti-Flag antibody. IgG was used as a negative control for the immunoprecipitation. **b** Coimmunoprecipitation of endogenous lamin A/C and Nestin from A549 and H1299 cell lysates with anti-lamin A/C antibody. **c** Immunofluorescence staining of Nestin (green) and lamin A/C (red) for their subcellular localization in lung cancer cells. Micrographs were obtained and analyzed by SIM. The images on the right and at the bottom highlight the co-localization of Nestin and lamin A/C. Scale bars, 5 μm. **d** Schematics of the utilized Myc-Nestin constructs. **e** Coimmunoprecipitation of endogenous lamin A/C and the indicated ectopically expressing constructs in HEK293T cells with anti-lamin A/C antibody. **f** qPCR analysis of Nestin expression in control and lamin A/C-knockdown cells. **g** Immunoblotting analysis of Nestin expression in control and lamin A/C-knockdown cells. For all quantifications, the data represented the mean ± SEM of three independent experiments, as assessed using unpaired t-test; ***P < 0.001

uncover the pathway responsible for this degradation of lamin A/C. Pretreatment with the caspase-6 inhibitor, Z-VEID-FMK, or with the autophagy inhibitor, 3-methyladenine (3-MA), yielded little recovery of lamin A/C in Nestin-knockdown cells (Fig. 5c). In contrast, the Nestin-knockdown-induced decrease in lamin A/C levels was entirely reversed by the broad-spectrum proteasome inhibitor, MG132 (Fig. 5d and Supplementary Fig. 5a), indicating that Nestin deficiency increases lamin A/C degradation through a proteasomal pathway. To further confirm this, we examined the levels of lamin A/C ubiquitination in Nestin-knockdown and control cells. We found that Nestin knockdown increased the ubiquitination of lamin A/C (Fig. 5e, f and Supplementary Fig. 5b, c), suggesting that Nestin negatively regulates the ubiquitination of lamin A/C in the tested cells. Furthermore, we performed cell fractionation experiments and found that Nestin knockdown induced the nuclear export of lamin A/C, and this could be reversed by the exportin 1-dependent nuclear export inhibitor, leptomycin B (LMB) (Fig. 5g and Supplementary Fig. 5d). Confocal microscopy revealed that lamin A/C principally localized at the nuclear lamina of control cells, whereas Nestin-knockdown

cells showed less lamin A/C at the nuclear lamina (Fig. 5h). In addition, Nestin knockdown increased the ubiquitination of lamin A/C in the cytoplasm, but not in the nucleus (Fig. 5i). These findings suggest that upon Nestin knockdown, lamin A/C undergoes nucleus-to-cytoplasm transport and is then subjected to proteolytic degradation in the cytoplasm. Thus, Nestin regulates the stability of lamin A/C by protecting it from proteasomal degradation.

**Nestin depletion triggers lamin A/C degradation in a Cdk5-dependent manner.** Since Nestin knockdown was associated with the dispersion of lamin A/C from the nuclear lamina, we hypothesized that lamin A/C may disassemble and be exported for degradation upon Nestin downregulation. We generated chimeric proteins consisting of A-type lamins fused to GFP and used fluorescence recovery after photobleaching (FRAP) analysis to investigate the dynamics of nuclear lamin A/C in living cells. The lamin A/C incorporated into the lamina is known to mobilize slowly[39,40]. However, we observed a clear and significant increase

in mobility upon Nestin knockdown, confirming that Nestin could contribute to the stability of the lamin A/C network in A549 and H1299 cells (Fig. 6a–c and Supplementary Fig. 6a–c). In contrast to B-type lamins, which are primarily found at the nuclear periphery, A-type lamins also localize in a mobile pool seen throughout the nuclear interior[41]. To further elucidate the Nestin-knockdown-induced changes in lamin A/C dynamics, we examined the amount of lamin A/C within soluble nuclear fraction (SNF) and extraction-resistant nuclear fraction (ERNF).

Nestin-knockdown cells exhibited decreases in ERNF-associated lamin A/C and increases in SNF-associated lamin A/C compared with controls (Supplementary Fig. 6d). Together, these results demonstrate that lamin A/C tends to disassemble and release itself from the nuclear envelope upon Nestin depletion.

Recent studies demonstrated that phosphorylate of A-type lamins by kinases, such as Cdk1, Cdk5, and Akt[42–45], may promote the disassembly and further export of the nuclear lamins. To further investigate the molecular mechanism

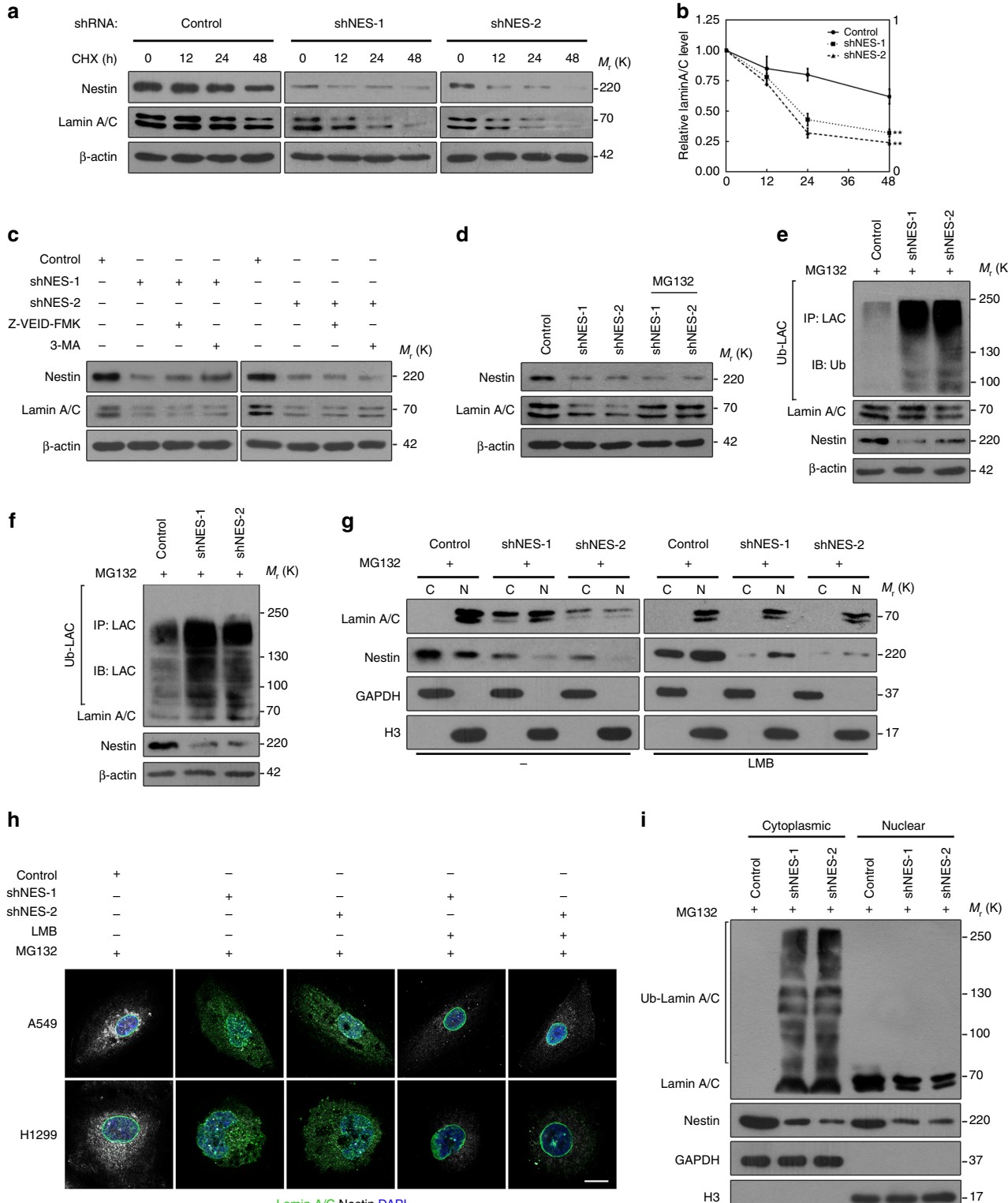

responsible for the disassembly of lamin A/C in Nestin-knockdown cancer cells, we treated Nestin-knockdown and control cells with inhibitors of Cdk1/2/5 and Akt, respectively. We found that inhibition of Cdk1/2/5 by roscovitine completely inhibited lamin A/C degradation, whereas the Cdk1/2 inhibitor, BMS-265246, and the Akt inhibitor, MK-2206, did not alter this disassembly status (Fig. 6d, e), suggesting Cdk5-mediated phosphorylation might be the candidate pathway for regulating lamin A/C disassembly in Nestin-knockdown cells. To test this hypothesis, we examined whether Nestin affects the expression of Cdk5. In Nestin-knockdown cells, we found that the protein levels of Cdk5 were unchanged, but its activity was significantly increased (Fig. 6f). Thus, Nestin knockdown appears to activate Cdk5, driving it to phosphorylate lamin A/C. Furthermore, Cdk5 reportedly phosphorylates lamin A/C at S22 and S392[43]. The phosphorylation levels of lamin A/C at S392 were greatly increased in Nestin-knockdown cells and restored by Cdk5 silencing, whereas no such changes were seen for S22 (Fig. 6g, h). Thus, Nestin knockdown led to the increased phosphorylation levels of lamin A/C at S392 in a Cdk5-dependent manner.

To further confirm the phenomena, we introduced a non-phosphorylatable (Ser to Ala) and a phosphomimetic (Ser to Asp) point mutation at S392 of lamin A/C. GFP-tagged wild-type (WT) lamin A/C and the above-mentioned two mutants were stably expressed in lung cancer cells. These cells were then transfected with control or Nestin shRNAs, and nuclear dispersion was analyzed by confocal microscopy. Exogenously expressed WT GFP-lamin A/C and GFP-lamin A/C-S392A localized predominantly to the lamina, with a small fraction residing in the nucleoplasm; robust dispersion of WT GFP-lamin A/C was observed with blurred nuclear rims following Nestin depletion. In contrast, there was little change in the distribution of GFP-lamin A/C-S392A (Supplementary Fig. 6e), indicating that this non-phosphorylatable lamin A/C mutant is resistant to Nestin knockdown inducing lamina dispersion. On the other hand, cells expressing GFP-lamin A/C-S392D showed intense lamina dispersion with no change in the protein levels of Nestin (Fig. 6i). These data, along with the results presented above, indicate that S392 is the major site for Cdk5-mediated phosphorylation of lamin A/C following Nestin depletion.

Taken together, our results indicate that Nestin regulates the stability of lamin A/C by protecting it from Cdk5-dependent proteasomal degradation, and Nestin deficiency can drive tumor cell senescence (Fig. 6j).

## Discussion

Nestin is widely expressed in numerous tumors and has been identified as a promising diagnostic and prognostic indicator.

Thus, elucidating the contributions of Nestin to cellular structure and function is essential for understanding the mechanisms underlying tumor pathogenesis and treatment. In this study, we revealed that Nestin played a key role in global nuclear organization. Mechanically, nuclear Nestin could form heteropolymers with lamin A/C, and that Nestin depletion increased the phosphorylation and subsequent degradation of lamin A/C in a Cdk5-dependent manner, which led to nuclear deformation and cellular senescence.

For a long time, lamins are considered as the only type IF proteins for nuclear localization and responsible for the formation of a dense meshwork of 10-nm filaments covering the inner surface of the nuclear envelope[22]. Other types of IFs are believed to reside and function exclusively in the cytoplasm. Until recently, the cytoskeleton protein keratin 17(K17) has been identified inside the nucleus through a classical bipartite NLS with a direct impact on cell proliferation and gene expression[46]. In the present study, we revealed that Nestin occurred in the natural context of tumor nucleus in culture and tissues in situ by microscopy, as well as subcellular fractionation. Because NLSs are necessary and sufficient to target a protein to the nucleus, we then verified that Nestin had a classical bipartite NLS (aa318–aa347) located closely downstream to its α-helical rod domain by truncation mutant analysis. In addition, GFP-Nestin fragment harboring three points substitution at vital basic amino-acid sites (K330A, R342A, and R343A) in the putative NLS showed attenuated ability to import to the nucleus, confirming that the identified NLS was sufficient for Nestin's nuclear transportation. However, compared with lamins, only a small portion of Nestin is present in the nucleus owing to the composition of its NLS, which has less concentrated basic amino acids and may be less efficient in driving Nestin to the nucleus. On the other hand, Nestin exhibits stronger nuclear localization ability than K17, whose NLS is half embedded in α-helical rod domain and may be masked or ineffective even in the setting of polymers[47]. These results indicate that the cytoplasmic non-lamin IF superfamily members may exhibit the nuclear localization ability, further clarifying the functional significance of nuclear-localized IFs will provide clues regarding the roles of IF proteins in physiological and pathological states.

Lately, the researchers had made great progress in understanding the functions of nuclear-localized IF proteins. For example, Hobbs et al.[46] showed that K17 promotes cell proliferation in skin tumor keratinocytes by regulating inflammatory gene expression[46]. What's more, Escobar-Hoyos et al.[48] demonstrated that K17 sustains cell cycle progression in cervical tumor epithelia by promoting the nuclear export of p27. These two independent studies demonstrated that IF proteins not only play fundamental roles for structure supporting but also participate in gene expression regulation. Interestingly, we found that Nestin

**Fig. 5** Nestin protects lamin A/C from proteasomal degradation. **a**, **b** Half-life analysis of lamin A/C in control and Nestin-knockdown A549 cells. All groups of A549 cells were treated with cycloheximide (50 μg/ml), harvested at the indicated times, and subjected to immunoblotting (**a**). Quantification of lamin A/C levels relative to β-actin is shown (**b**). **c**, **d** Identification of the pathway responsible for lamin A/C degradation upon Nestin knockdown. Control and Nestin-knockdown A549 cells were treated with Z-VEID-FMK (20 μM, 2 h), 3-MA (5 mM, 2 h) (**c**), or MG132 (20 μM, 6 h) (**d**), and then proteins were extracted and subjected to immunoblotting. **e**, **f** The effects of Nestin knockdown on ubiquitination of lamin A/C were analyzed by in vivo ubiquitination assays. Control and Nestin-knockdown A549 cells were treated with MG132 (20 μM) for 6 h before harvest. Lamin A/C was immunoprecipitated with anti-lamin A/C antibody and immunoblotted with anti-Ub antibody (**e**) or anti-lamin A/C antibody (**f**). **g**, **h** Immunoblotting and immunostaining analysis of lamin A/C distribution in nucleus and cytoplasm. Control and Nestin-knockdown A549 cells were treated with MG132 (20 μM, 2 h) and then treated with or without 25 ng/ml LMB. After 4 h, the cells were fractionated for immunoblotting analysis (C cytoplasmic cell lysates, N nuclear cell lysates) (**g**) or stained with anti-lamin A/C antibody (green) and anti-Nestin antibody (white). Scale bars, 20 μm (**h**). **i** The effects of Nestin knockdown on ubiquitination of cytoplasmic lamin A/C were analyzed by in vivo ubiquitination assays. Control and Nestin-knockdown A549 cells were treated with MG132 (20 μM) for 6 h, harvested, and fractionated, and the cytoplasmic or nuclear lamin A/C fractions were immunoprecipitated and immunoblotted with anti-lamin A/C antibody. The quantified results were presented as mean ± SEM of three independent experiments, as assessed using two-way ANOVA test. **$P < 0.01$

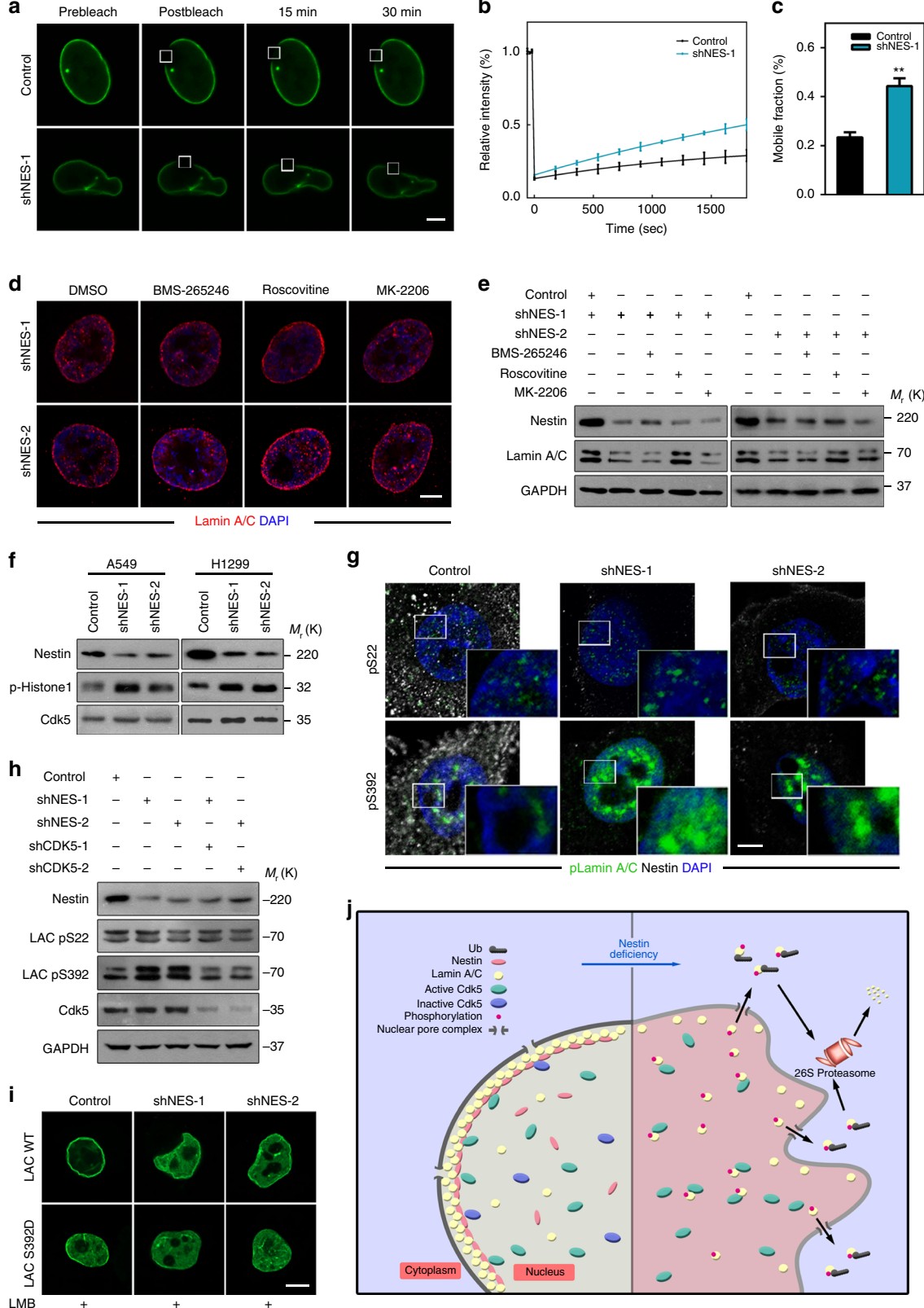

knockdown significantly triggered nuclear deformation. Because lamins are major structural components of the nucleus lying beneath the INM[22] and defects in nuclear lamins could cause nuclear deformation[49–51], we then speculate whether Nestin participates in the regulation of lamina stability. Consistently, immunoprecipitation assays and domain-mapping studies revealed that the C terminus of the Nestin rod domain (aa182–aa313) was critical for its interaction with lamin A/C, and downregulation of Nestin resulted in lamin A/C dispersion and degradation and then tumor cell senescence. Previous studies have demonstrated that tumor cells can be readily induced to undergo senescence by genetic manipulations or by treatment

**Fig. 6** Phosphorylation by Cdk5 drives the degradation of lamin A/C after Nestin knockdown. **a**, **b** FRAP anaysis of GFP-lamin A/C in control and Nestin-knockdown A549 cells. **a** Representative fluorescence images showed (from left to right): pre-bleach image; image taken immediately after bleaching; and two frames (15 and 30 min) from a time-lapse image series taken at 20-s intervals over half an hour. Scale bars, 5 μm. **b** The mean normalized FRAP data measured from the bleached regions ($n = 5$). **c** The average mobile fractions of the lamin A/C proteins were calculated from the FRAP data shown in **b** ($n = 5$). **d**, **e** Immunostaining and immunoblotting analysis of lamin A/C status. A549 cells of control or Nestin-knockdown were treated with BMS-265246 (1 μM), roscovitine (20 μM), or MK-2206 (3 μM). After 24 h, cells were stained with anti-lamin A/C antibody (red) and DAPI (blue) (Scale bars, 5 μm) (**d**) or immunoblotted with the indicated antibodies (**e**). **f** The effects of Nestin knockdown on Cdk5. Cdk5 was immunoprecipitated in control or Nestin-knockdown A549 and H1299 cells, and kinase assays were conducted using a Cdk5 substrate peptide, as described in the Methods section. **g** Representative immunofluorescence data for pS22 lamin A/C or pS392 lamin A/C (green) plus Nestin (white) in control and Nestin-knockdown A549 cells. Scale bars, 5 μm. **h** Immunoblotting analysis of lamin A/C phosphorylation levels affected by Nestin and Cdk5 knockdown. A549 cells were transfected with the indicated shRNAs, and proteins were extracted and subjected to immunoblotting. **i** Confocal microscopy was employed to determine the subcellular localization of WT GFP-lamin A/C, compared with GFP-lamin A/C-S392D. A549 cells transfected with the indicated constructs were treated with 25 ng/ml leptomycin B (LMB) for 6 h, and then harvested, fixed, and stained with DAPI (blue). Scale bars, 10 μm. **j** Illustration of the Nestin–Cdk5–lamin A/C axis in the regulation of tumor senescence. The quantified results were presented as mean ± SEM of five independent experiments (**b**, **c**), as assessed using unpaired $t$-test (**c**). **P < 0.01

with chemotherapeutic drugs, radiation, or differentiating agents[52,53]. Thus, Nestin deficiency inducing tumor senescence indicates Nestin might be a therapeutic target for tumor treatment. Most recently, Li et al.[54] showed that Nestin expression is gradually loss in the primary spongiosa of long bones during late puberty, leading to normal programmed senescence. Whether Nestin affects replicative senescence of normal cells still need to be clarified.

Posttranslational modifications (PTMs) play a central role in regulating the functional properties of IFs[55]. Previous studies found that high levels of phosphorylation was associated with the disassembly and degradation of IFs[43,44]. During mitosis, S392 and S22 are phosphorylated for lamin A/C filaments disassembly, though S392 is less important than the S22 site, with respect to assembly[39]. However, our results showed that the phosphorylation levels of lamin A/C at S22 had little change during Nestin knockdown. Thereby, the results above suggested that Nestin has a specific impact on the PTM of lamin A/C at S392. Moreover, the phosphorylation levels of lamin A/C at S392 were restored by Cdk5 silencing in Nestin-knockdown cells, indicating that Nestin deficiency led to the increased phosphorylation levels of lamin A/C at S392 in a Cdk5-dependent manner. In addition, It is well demonstrated that Nestin has a bidirectional crosstalk with Cdk5. On one hand, the Nestin filament reorganization during differentiation is dependent on Cdk5-mediated Nestin phosphorylation[56]. On the other hand, Nestin binds to Cdk5/p35 and stabilizes p35. Downregulation of Nestin increases the calpain cleavage of p35 to p25 and Cdk5/p25 activity, which is linked to neuronal cell death[57]. Similarly, our data showed that loss of Nestin resulted in the activation of Cdk5. Subsequently, activated Cdk5 phosphorylated lamin A/C at S392 and contributed to lamina dispersion.

In conclusion, Nestin has now been found to occur and function inside the nucleus. The available evidence suggests that the nuclear Nestin regulates the homeostasis of nuclear lamina and involves in the tumor cellular senescence.

## Methods

**Ethics statement**. BALB/c nude mice were purchased from Beijing Vital River Laboratory. Animal protocols were approved by the Ethical Committee of Sun Yat-sen University. The clinical lung cancer samples used in this study were obtained from patients treated in the Department of Thoracic Surgery from the First Affiliated Hospital of Sun Yat-sen University. For the research use of all clinical samples, informed consent was obtained in writing. This study was approved by the Committees for Ethical Review of Research involving Human Subjects of Sun Yat-Sen University.

**Cell culture and tumor inoculation**. H1299, A549, LN229, C4-2, PC-3, and HEK293T cells were purchased from American Type Culture Collection; U251,

TE-1, and Eca-109 were purchased from the Chinese Academy of Sciences (Shanghai, China). All cell lines were cultured in Dulbecco's modified Eagle's medium (DMEM)/high glucose (Hyclone) containing 10% fetal bovine serum (FBS, GIBCO). All cell lines have been tested negative for mycoplasma contamination. For mouse xenograft models, animals were randomly allocated into groups receiving cell line injections. A549 and H1299 cells ($2 \times 10^6$) were suspended in 100 μl phosphate-buffered saline (PBS) and injected subcutaneously into the left flanks, respectively, of 8-week-old male BALB/c nude mice ($n = 3$ mice/group). After about 3 weeks, subcutaneous tumors were collected and immunofluorescence staining was used to examine the localization of Nestin. No method of blinding was used.

**Vectors and reagents**. For knockdown of Nestin expression, retrovirus vectors (pSM2) encoding shRNAs were purchased from Open Biosystems (Huntsville, AL, USA). Oligonucleotides for shRNA-mediated targeting of human lamin A/C and Cdk5 were cloned into pLL3.7 (Addgene). The target sequences of shRNAs against Nestin, lamin A/C (LAC), and Cdk5 were listed in Supplementary Table 1. Full-length Nestin and its deletion mutants were cloned into the pcDNA3.1-Myc vector (Invitrogen). DsRed, Flag-tagged Nestin, and the GFP-tagged Nestin truncation mutants were constructed using Invitrogen's Gateway System. pBABE-H2B-GFP (26790) and pBABE-puro-GFP-WT-lamin A (17662) were obtained from Addgene. WT GFP-lamin A/C and GFP-lamin A/C-S392D were provided by Dr. J.E. Eriksson (Åbo Akademi University, Turku, Finland). GFP-Lamin A/C-S392A mutant was constructed based on overlap extension PCR. BMS-265246, roscovitine, MK-2206, and 3-MA were purchased from Selleck. Z-VEID-FMK and LMB were purchased from Calbiochem. MG132 and CHX were purchased from Sigma.

**Cell proliferation and nuclear shape measurements**. Cells were transfected with shNestin or control shRNA by a lentivirus system for 36 h, seeded to 12-well plates ($5 \times 10^4$ cells/well) and incubated for 4 days. Cells were counted at the indicated times, and growth curves were plotted. For establishing dual-color cells, A549 and H1299 cells were incubated with a 1:1 mixture of RFP lentivirus and H2B-GFP retrovirus. After 12 h incubation, medium was replaced with DMEM/high glucose containing 10% FBS. Three days later, cells expressing RFP and H2B-GFP were sorted using an Influx Cell Sorter (BD, USA) and cultured on 12-well plates. The CellProfiler software was used to quantify nuclear circularity and nuclear area from H2B-GFP expressing or 4′,6-diamidino-2-phenylindole staining grayscale pictures. Shape features of nuclei were determined using the "identify primary objects" followed by the "measure object size shape" and "export to spread sheet" module. Among these features, we calculated the nuclear circularity or contour ratio as: $4\pi \times$ area perimeter$^{-2}$. (The contour ratio has a maximum value of 1 for a circle and decreases as the nuclear shape becomes increasingly convoluted)[28,30–32].

**SA-β-gal staining**. SA-β-gal staining was performed using a senescence-associated β-galactosidase kit (Beyotime, C0602). A549 and H1299 cells seeded in 12-well plates were washed twice with PBS, fixed in fixative for 15 min, washed, and incubated overnight at 37 °C with the working solution of β-galactosidase plus X-Gal. The senescent cells were observed under an optical microscope (leika, DMi8) and counted from three random fields of vision.

**Flow cytometric analysis of cell cycle and apoptosis**. For flow cytometry-based cell cycle analysis, A549 and H1299 cells were harvested, washed, resuspended in PBS, fixed in ethanol, and stained with propidium iodide (PI; Vazyme, A211-02). Cellular DNA contents were measured using a FACS Calibur flow cytometer (Beckman Coulter, USA). The profiles of cells in G0/G1, S, and G2/M were determined. For flow cytometry-based apoptosis analysis, cells were collected, washed twice with PBS, resuspended in 100 μl 1× binding buffer, and stained with

Annexin V and PI (Vazyme, A211-02), according to the manufacturer's protocol. After 15 min of incubation at room temperature in the dark, cells were analyzed by flow cytometry (Beckman Coulter, USA). At least 20 000 cells were collected for each sample.

**Gene targeting by the CRISPR/Cas9 system**. The first coding exon of Nestin was selected for gRNA design. CRISPRs were designed at http://crispr.mit.edu, as provided by the Zhang laboratory and cloned into the pX458 CRISPR/Cas9 vector (Addgene). The sequences used to clone the G1, G2, and G3 gRNAs were presented in Supplementary Figure 2. Two days after H1299 cells were transfected with control (Cas9-GFP) and Nestin-knockout (G2-Cas9-GFP) plasmids, those expressing GFP were sorted using an Influx Cell Sorter (BD, USA) and cultured on 12-well plates. Twenty-four hours later, half of the Nestin-knockout cells were transfected with GFP-lamin A/C. The cells were then cultured for 3 days and harvested for further experiments.

**T7E1 assay**. PCR amplicons from selected genomic region were purified with a PCR Purification Kit (Tiangen, DP214-03), denatured, and annealed in NEBuffer 2 (NEB) using a thermocycler. Hybridized PCR products were digested with T7E1 (NEB, M0302L) for 30 min at 37 °C and subjected to 2% agarose gel electrophoresis. The primers used for the PCR amplification of Nestin in this case were listed in Supplementary Table 2.

**Quantitative PCR (qPCR)**. Total RNA was prepared using the TRIzol reagent (Molecular Research Center, Inc.) according to the manufacturer's instructions and 1 μg was subjected to reverse transcription using a RevertAid First Strand cDNA Synthesis Kit (Thermo, K1622). The obtained cDNAs were subjected to real-time PCR analysis using the FastStart Essential DNA Green Master Mix (Roche, 06924204001), and primers specific to *GAPDH*, *NES*, *LMNA*, *LMNB1*, *IL6*, and *IL8*. Samples were run in triplicate and normalized to GAPDH. Relative expression was calculated using the comparative $C_T$ ($\Delta\Delta C_T$) method[58]. The designed PCR primers were listed in Supplementary Table 2.

**Cell fractionation**. Cytoplasmic and nuclear fractions were separated with a Nucleoprotein Extraction Kit (Sangon Biotech, C510001). For more elaborate subcellular fractionation, the nuclear pellet was resuspended in ice-cold 20 mM Hepes (pH 7.9), 0.4 M NaCl, 1 mM EDTA, 1 mM EGTA, 1 mM dithiothreitol, and 1 mM phenylmethylsulfonyl fluoride, and agitated at 4 °C for 15 min. The nuclear lysate was centrifuged for 30 min at 4 °C to obtain the SNF and the pellet containing the ERNF[59].

**Coimmunoprecipitation and immunoblotting**. For immunoprecipitation assays, cells were lysed using Pierce IP lysis buffer (ThermoFisher Scientific, 87787) supplemented with protease inhibitor cocktail (Roche, 04693116001) and phosphatase inhibitor cocktail (Roche, 04906837001). The cell lysates were centrifuged, and then immunoprecipitated overnight at 4 °C using the indicated primary antibodies followed by incubation with Dynabeads Protein G (Life Technologies, 10003D) for 1 h. The immunocomplexes were washed twice with IP lysis buffer before being resolved by SDS-polyacrylamide gel electrophoresis (SDS-PAGE) and immunoblotted with indicated antibodies. For immunoblotting, cells were washed with PBS, lysed on ice in RIPA buffer (Millipore, 20-188) supplemented with protease inhibitor cocktail (Roche, 04693116001) and phosphatase inhibitor cocktail (Roche, 04906837001), and resolved by SDS-PAGE with corresponding concentration and immunoblotted with indicated antibodies. All antibodies for normal immunoblotting and immunoprecipitation were listed in Supplementary Table 3. All immunoblotting experiments were performed at least three times, and representative data were shown. Images of uncropped immunoblots are shown in Supplementary Fig. 7.

**In vivo ubiquitylation assays**. For detection of endogenous ubiquitylation of lamin A/C, transfected A549 and H1299 cells were treated for 6 h with a proteasome inhibitor (MG132, 20 μM), and the cells were lysed in IP lysis buffer (ThermoFisher Scientific, 87787) supplemented with protease inhibitor cocktail (Roche, 04693116001) and phosphatase inhibitor cocktail (Roche, 04906837001). Extracts were immunoprecipitated with anti-lamin A/C antibody followed by immunoblotting with anti-Ub or anti-lamin A/C antibody.

**Cdk5 activity assay**. Following Cdk5 immunoprecipitation, immunoprecipitates were washed three times with lysis buffer and once with kinase buffer (CST, 9802). The washed beads were then incubated with kinase buffer containing 2.5 μg of histone H1 (Calbiochem, 382150) and 0.2 mM unlabeled ATP (CST, 9804) in a final volume of 25 ml at 30 °C for 10 min. The reactions were then stopped by boiling with Laemmli sample buffer for 5 min. The supernatant was analyzed by immunoblotting.

**SIM experiments**. Cells were cultured on coverslips to the appropriate density, fixed with 4% paraformaldehyde for 15 min, and permeabilized with 0.2% Triton

X-100. The slides were then blocked in 1% bovine serum albumin (BSA) for 30 min, incubated with the appropriate primary antibodies diluted in 1 % BSA for 1.5 h, washed with PBS, and incubated with secondary antibodies (1:500 dilution) for 1 h. The slides were then washed, mounted, and observed by Nikon N-SIM microscopy. For confocal images, Zeiss 780 and 800 confocal microscopies were employed. All antibodies for immunofluorescence staining were listed in Table S3.

**Immunohistochemistry (IHC)**. Paraffin-embedded lung cancer tissue sections (3 μm) were subjected to immunostaining using an UltraSensitive™ SP (Mouse/Rabbit) IHC Kit (MXB, KIT-9710). Each section was deparaffinized, treated with 3% $H_2O_2$ for 15 min, microwaved in 10 mM citric sodium (pH 6.0) for 15 min, incubated with an anti-Nestin antibody (Abcam, ab27952, 1:200) overnight at 4 °C, and then incubated with a secondary antibody for 30 min at 37 °C. Signal amplification and detection was performed using the DAB system according to the manufacturer's instructions (MXB, MAX-001). The slides of A549 and H1299 cells were fixed with 4% paraformaldehyde and permeabilized with 0.5% Triton X-100, followed by 3% $H_2O_2$ incubation and later steps as described above. All antibodies for immunohistochemistry are listed in Supplementary Table 3.

**Fluorescence recovery after photobleaching**. Cells expressing GFP-lamin A/C were scanned using laser at low laser power by Nikon A1 confocal microscopy. Three pre-bleach images were recorded (1–2% 488 laser line; 2 μm × 2 μm confocal section), and then a selected area was bleached using high laser power (full power, 488 laser line). FRAP was recorded using time-lapse images of the original section and quantitative image analysis of the bleached zone, as performed using Nikon software. The raw intensity data obtained from the recovery phase were normalized by subtracting the background fluorescence to compensate for laser fluctuations. The mobile fractions of the GFP-lamin A/C constructs were calculated by curve-fitting the fluorescence recovery data with the following formula to estimate one-phase disassociation–association kinetics[39]: $I_{(t)} = I + P(1 - \exp^{-kt})$. $I_{(t)}$: normalized fluorescence intensity in the bleached area at a time $t$; $I$: normalized initial fluorescence intensity in the bleached area immediately after photobleaching; $P$: normalized fluorescence at plateau; $k$: constant.

**Statistical analysis**. Results were reported as the mean ± SEM of at least three independent experiments. Sample sizes were indicated in the figure legends. Comparisons were performed with a two-tailed unpaired Student's *t*-test or two-way ANOVA (for multi-group comparisons). *P* values less than 0.05 were considered significant, and the level of significance is indicated as \**P* < 0.05, \*\**P* < 0.01, and \*\*\**P* < 0.001.

## Data availability

The data that support the findings of this study are available within the article and its supplementary Files or available from the corresponding authors on reasonable request.

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

## Acknowledgements

We are very grateful to J.E. Eriksson (Åbo Akademi University, Turku, Finland) for providing the WT GFP-lamin A/C and GFP-lamin A/C-S392D expression plasmids; B.T. Lahn (University of Chicago, Chicago, USA) and K. Zhang (University of California, San Diego, USA) for providing advice. This work was supported by the National Key Research and Development Program of China, Stem cell and Translational Research (2018YFA0107203, 2017YFA0103403, and 2017YFA0103802), Strategic Priority Research Program of the Chinese Academy of Sciences (XDA16010102), the National Natural Science Foundation of China (81425016, 81730005, and 31771616), the Natural Science Foundation of Guangdong Province (S2013030013305 and 2017A030310237), Frontier and Innovation of Key Technology Project in Science and Technology Department of Guangdong Province (2014B020226002, 2015B020228001, 2015B020229001, 2016B030229002, 2016B030230001, and 2017B020231001), Key Scientific and Technological Program of Guangzhou City (201803040011 and 201704020223), Wenzhou Committee of Science and Technology of China (ZS2017008), Guangdong Province Universities and Colleges Pearl River Scholar Funded Scheme (GDUPS, 2013), and China Postdoctoral Science Foundation (2016M602583 and 2017T100657).

## Author contributions

Y.Z. and J.W. designed and performed experiments, analyzed data, and wrote the manuscript. W.H. performed bioinformatics analysis and wrote the manuscript. J.C., J.B. and Q.K. designed the experiments and interpreted data; Y.W. and X.L. processed the images from experiments. Y.H., Y.Q. and Q.L. assisted in some in vitro experiments; X.S.

and Y.S. assisted in the mouse experiments; T.W., H.S., Y.G. and Y.Z. edited the manuscript. A.P.X., M.W. and Y.C. supervised the project and wrote the manuscript.

## Additional information

**Competing interests:** The authors declare no competing interests.

