## [Peer Review File · Nature Communications]

Reviewer #1, Expertise: Nestin, Cancer (Remarks to the Author):

The authors demonstrate that nestin can reside in the nucleus and protect cancer cells from senescence and claim that the findings establish a new role for nestin in tumor senescence, which involves its nucleus-localized form and interaction with lamin A/C. The authors observations are interesting but the manuscript would have been much stronger using cells in which the nucleus is labeled with GFP and the cytoplasm with RFP to image nuclear movement and deformation (Cancer Research 64, 4251-4256, 2004; Cancer Res. 65, 4246-4252, 2005; Nature Protocols 1, 775-782, 2006). The authors also should mention that nestin plays a critical role in hair follicle-associated-pluripotent (HAP) stem cells that differentiate into many other cell types (Proc. Natl. Acad. Sci. USA 100, 9958-9961, 2003; Proc. Natl. Acad. Sci. USA 102, 5530-5534, 2005; Cell Cycle 14, 2362-2366, 2015).

Reviewer #2, Expertise: senescence, lamins (Remarks to the Author):

This is an elegant study. The authors demonstrate the knock down of nestin causes cdk5 dependent dispersal and degradation of lamin A/C via the proteasome. I have the following comments that should be addressed prior to publication.

Major points.

1. The authors describe a senescence-like phenotype on knock down of nestin. But the cell population is still proliferating. Are the senescence markers expressed in a subset of senescent cells while other cells continue to proliferate, or are the senescence markers expressed uniformly in a population of slowly proliferating cells? The authors should do long term EdU label and IF to ask whether there is a subset of non-proliferating cells that express the senescence markers.

Also re. the senescence phenotype, what is the p53 and pRB status of these cells? They upregulate p16 and p21 on nestin knock down. However, increased p16 will have no effect in pRB null cells. Clearly, nestin knock down has a dramatic phenotype in these cells, but the authors should consider the phenotype and genetic context of the cells to know whether this is a canonical senescence program or something related but different.

Cell senescence is best characterized and understand in primary, non-transformed cells. If the authors really think this is a senescent state, they should ask whether similar happens in primary human cells. Otherwise, they should moderate their discussion of senescence – it may be simply

more of a cell stress response with features of senescence. There are differences between this phenotype described here and canonical senescence in primary cells - in the latter lamin B1 downregulation is a hallmark of senescence.

2. Figure 6. Have the authors tested a non-phosphorylatable lamin A/C mutant? This should be resistant to nestin knock down.

Minor points

1. Why did the authors examine lung cancer cell lines? In the introduction, the authors cite glioma, melanoma,

gastrointestinal stromal tumors and prostate cancer as showing elevated nestin. Is there evidence for recurrent overexpression/alteration in lung?

2. The validity of the nestin in the nucleus should be confirmed with a 2nd antibody for IHC and knock down for immunofluorescence and subcellular fractionation (Figure 5g shows knock down combined with subcellular fractionation).

3. Figure 4A is lacking a key control – omit the flag-tagged nestin from the anti-flag IP.

4. Figure 4b. Can the authors confirm the lamin A/C – nestin interaction by IP of endogenous nestin?

5. Figure 4d. Is the C terminus of the rod domain sufficient for interaction with nestin, or only necessary?

Reviewer #3, Expertise: nuclear organisation, super resolution microscopy and FRAP (Remarks to the Author):

In this manuscript, Zhang et al. set out to study subcellular locations and interactions with lamin A/C for Nestin. They have listed a few new findings: first, classical nuclear localization signals were found at the Nestin rod domain. Second, the rod domain can directly interact with lamin A/C. Third, the lack of Nestin causes the phosphorylation and nuclear export of lamin A/C.

There are some major and minor issues as follows that need to be addressed before a recommendation of publication can be made:

1) In the introduction, the authors just simply listed previous findings, which makes the reading experience not very enjoyable. This reviewer suggests that it could be better organized by classifying these findings with a sound and concise flow.

2) Circularity of nuclear shape is used as an important and sole indicator for quantitatively distinguishing the normal and abnormal nuclear shapes with or without knockdown of Nestin in different cell lines. Although a previous publication was referenced and a few words were provided in Methods, a more detailed introduction is still needed to clarify the data process. Also, if this is a well-accepted approach, this reviewer is curious why only one previous publication was cited; if not, a serious question is how reliable the approach is. If this question cannot be well addressed, the rest studies in the manuscript would start shaking.

3) Another serious concern is about the NLS-dependent nuclear-cytoplasmic localization of Nestin. There are several confusing aspects. First, the information of software and data process used to predict weak or strong NLSs is insufficient to know if a score of ~ 4.2 means weak or strong signals. Second, the authors named all signals (NLS1, NLS2, NLS3) as NLSs, but the tests on NLS1 and NLS3 indicated that these signals caused Nestin to stay primarily in cytoplasm. So, these signals might be NESs rather than NLSs? Finally, an accurate and standard approach is to use Cytoplasm/Nucleus intensity ratio of GFP-tagged proteins to quantitatively determine their subcellular locations. For example, it's hard to tell how big the difference is between F4 and F5. Also, there is no clue how a conclusion that Nestin only stays at NE for F2 test was drew.

4) Finally, this reviewer thinks that the section of discussion could be better handled by concisely highlighting the major results and their impacts on the specific field. The current version is too sparse and unclear.

Response to the reviewers

We appreciate the reviewers for their constructive feedback. To address the concerns raised by the reviewers, we performed additional experiments as well as implementing considerable changes to the manuscript. As a result, we believe the manuscript is much stronger. Below, we list the reviewers' comments, and describe point-by-point how we have addressed them.

Reviewer #1:

Major Point: The authors demonstrate that nestin can reside in the nucleus and protect cancer cells from senescence and claim that the findings establish a new role for nestin in tumor senescence, which involves its nucleus-localized form and interaction with lamin A/C. The authors observations are interesting but the manuscript would have been much stronger using cells in which the nucleus is labeled with GFP and the cytoplasm with RFP to image nuclear movement and deformation (Cancer Research 64, 4251-4256, 2004; Cancer Res. 65, 4246-4252, 2005; Nature Protocols 1, 775-782, 2006). The authors also should mention that nestin plays a critical role in hair follicle-associated-pluripotent (HAP) stem cells that differentiate into many other cell types (Proc. Natl. Acad. Sci. USA 100, 9958-9961, 2003; Proc. Natl. Acad. Sci. USA 102, 5530-5534, 2005; Cell Cycle 14, 2362-2366, 2015).

Response: We are grateful for your kind advice. This method, dual-color fluorescent cells with one color in the nucleus and the other in the cytoplasm, indeed enables nuclear dynamics to be visualized in living cells in vitro. As suggested, we performed additional experiments to image nuclear movement and deformation. To obtain the dual-color cells, red fluorescent protein (RFP) was expressed in the cytoplasm of A549 and H1299 cell lines, and green fluorescent protein (GFP) linked to histone H2B was expressed in the nucleus according to the well-established method (*Cancer Res.* 2004, 64: 4251-4256; *Cancer Res.* 2005, 65: 4246-4252; *Nat Protoc.* 2006, 1: 775-782). Consistent with our previous observation, we found that Nestin-knockdown cells exhibited significant nuclear malformations with decreased mean circularity and increased fraction of abnormally shaped nuclei compared with control cells. **The corresponding data was added as Figure 1a-1c.**

In addition, the reviewer mentioned that Nestin plays a critical role in hair follicle-associated-pluripotent (HAP) stem cells that differentiate into many other cell types (*Proc Natl Acad Sci U S A.* 2003; *Proc Natl Acad Sci U S A.* 2005; *Cell Cycle.* 2015). **In the revised manuscript, we added the related references in the "Introduction" section.**

Reviewer #2:

Major Point 1: The authors describe a senescence-like phenotype on knock down of nestin. But the cell population is still proliferating. Are the senescence markers expressed in a subset of senescent cells while other cells continue to proliferate, or are the senescence markers expressed uniformly in a population of slowly proliferating cells? The authors should do long term EdU label and IF to ask whether there is a subset of non-proliferating cells that express the senescence markers.

Response: We appreciate the helpful comments and advice of the reviewer. As suggested, we performed additional experiments to investigate whether there was a subset of non-proliferating cells that expressed the senescence marker (SA- β -gal). Based on 24 hr EdU labeling of cultured cells, we observed the proliferation marker and senescence marker were barely co-expressed in one cell. Instead, both control and Nestin-knockdown cells mainly consist of three subsets: proliferating cell population (EdU⁺/SA- β -gal⁻), senescent cell population (EdU⁻/SA- β -gal⁺) and quiescent cell population (EdU⁻/SA- β -gal⁻). As shown in Attached/Reviewer Figure 1a, Nestin knockdown significantly up-regulated the percentage of senescent cell population, which is consistent with our previous observation in Figure 1e and 1f. In addition, Nestin knockdown significantly decreased the percentage of proliferating cell population.

However, it is worth noting that there were still some EdU⁺/SA- β -gal⁻ cells (~16%) in Nestin-knockdown group. We speculate incomplete Nestin depletion by RNA interference method cause the residual proliferating cell subset (EdU positive cells).

To further confirm the phenomena that Nestin deficiency cause cell senescence, we

ablated Nestin expression in the A549 and H1299 cell line with CRISPR/Cas9 gene editing system. More specifically, we transfected A549 and H1299 cells with control (Cas9-GFP) and Nestin-knockout (G2-Cas9-GFP) plasmids. Two days later, those expressing GFP were sorted using an Influx Cell Sorter (BD), seeded to 24-well plates (3×10^4 cells/well) and incubated for 3 days. Cells were counted at the indicated times, and growth curves were plotted. As shown in Attached/Reviewer Figure 1b, the Nestin-knockout cell population is **no longer proliferating** since the second day seeded on 24-well plates when compared to the control group. Therefore, we suspect that the residual proliferating cell population in Nestin-knockdown group is due to incomplete Nestin depletion.

Attached/Reviewer Figure 1 The effects of Nestin on proliferation and senescence.

(a) Left: long term EdU label and SA- β -gal staining of A549 and H1299 cells. Arrows

indicate the residual EdU positive cells in Nestin-knockdown group. Right: corresponding quantification in figure a. Scale bars, 60 μ m.

(b) Proliferation of A549 and H1299 cells, as detected by a cell-counting assay.

Major Point 2: Also, re. the senescence phenotype, what is the p53 and pRB status of these cells? They upregulate p16 and p21 on nestin knock down. However, increased p16 will have no effect in pRB null cells. Clearly, nestin knock down has a dramatic phenotype in these cells, but the authors should consider the phenotype and genetic context of the cells to know whether this is a canonical senescence program or something related but different.

Response: This is a good point. Induction of senescence is thought to involve the p53-p21^{CIP1} and p16^{INK4A}-retinoblastoma protein (Rb) pathways, (*Curr Opin Genet Dev.* 2003, 13: 77-83). p53 effectively blocks cell cycle progression by upregulating its transcriptional target, p21^{CIP1} (*Cell.* 2009, 137: 413–431). As suggested, we detect the p53 level in A549 (wt p53) and H1299 (p53 null) cells and find that Nestin knockdown significantly up-regulate p53 and p21^{CIP1} expression in A549 cells. On the other hand, we do not detect the p53 protein expression in H1299 cells, which is similar with the previous studies (*Biochem Biophys Res Commun.* 2002, 293: 1248-1253; *Mol Cancer.* 2010, 9: 220). Therefore, it implies that Nestin knockdown inducing p21 expression is independent of p53 pathway, might be regulated by AP-1 (*Biochem Biophys Res Commun.* 2002, 293: 1248-1253) or FMN2 (*Mol Cell.* 2013, 49: 922-33), in H1299 cells.

As suggested, we also detect the status of pRb in these two lung cancer cell lines. The results show that Nestin knockdown decrease the phosphorylation levels of pRb in both A549 and H1299 cells, suggesting that hypophosphorylated pRb activation participate in the Nestin knockdown inducing cell cycle arrest. **As suggested, we added the corresponding data in Supplementary Figure 1i.**

Major Point 3: Cell senescence is best characterized and understand in primary, non-transformed cells. If the authors really think this is a senescent state, they should

ask whether similar happens in primary human cells. Otherwise, they should moderate their discussion of senescence – it may be simply more of a cell stress response with features of senescence. There are differences between this phenotype described here and canonical senescence in primary cells - in the latter lamin B1 downregulation is a hallmark of senescence.

Response: We appreciate the professional comments of the reviewer. Actually, in the adult, Nestin-expressing cells are found frequently in areas of stem/progenitor populations. Once these cells become differentiated and cease to divide, Nestin expression is downregulated (*Cell Mol Life Sci.* 2018, 75: 2177-2195). The previous report also demonstrates that the primary lung epithelial cells do not express Nestin, and the reexpression of Nestin is observed in neoplastic transformation. (*Cancer Sci.* 2015, 106: 803-811). Therefore, we could not perform the experiments to study the effect of Nestin on senescence in primary lung epithelial cells. For reviewer's concern, normal cell senescence is broadly defined as the physiological program of terminal growth arrest, which can be triggered by alterations of telomeres or by different forms of stress. However, tumor cells can be readily induced to undergo senescence **by genetic manipulations** or **by treatment** with chemotherapeutic drugs, radiation, or differentiating agents (*Proc Natl Acad Sci U S A.* 2007, 104: 13028-13033; *Nat Rev Cancer.* 2011, 11: 503-511). As reviewer's comment, **treatment-induced senescence**, which has both similarities with, and differences from, **replicative senescence** of normal cells, was shown to be one of the key determinants of tumor response to therapy in vitro and in vivo (*Cancer Res.* 2003, 63: 2705-2715; *Trends Cell Biol.* 2012, 22: 211-219). In this manuscript, we observed Nestin knockdown induced cell cycle inhibitors and increased SA- β -gal activity, which accords with the features of **treatment-induced senescence**.

Accordingly, we agree that this is a good idea and revise the discussion to make this point more clear.

Major Point 4: Figure 6. Have the authors tested a non-phosphorylatable lamin A/C mutant? This should be resistant to nestin knock down.

Response: As suggested, we constructed a non-phosphorylatable lamin A/C mutant vector through introducing a Ser to Ala point mutation at S392 of lamin A/C (lamin A/C^{S392A}). Briefly, we amplified S392A-lamin A/C DNA fragment by overlapping extension PCR from GFP-lamin A/C pEGFP-C1 (template), digested it with *Ava*I and then ligated it with *Ava*I-digested template vector DNA. To test the effect of this non-phosphorylatable mutant, we expressed GFP-tagged wild-type (WT) lamin A/C and lamin A/C^{S392A} in A549 and H1299 cells and observed the nuclear dispersion by confocal microscopy. The results showed that exogenously expressed WT and S392A lamin A/C localized predominantly to the lamina. Following Nestin depletion, cells expressing WT lamin A/C showed robust lamina dispersion. In contrast, there was little change in the distribution of lamin A/C^{S392A}. Taken together, our results indicate that this non-phosphorylatable lamin A/C mutant is resistant to Nestin knockdown inducing lamina dispersion. **In the revised manuscript, we added the corresponding data to Supplementary Figure 6e.**

Minor points 1: Why did the authors examine lung cancer cell lines? In the introduction, the authors cite glioma, melanoma, gastrointestinal stromal tumors and prostate cancer as showing elevated nestin. Is there evidence for recurrent overexpression/alteration in lung?

Response: We appreciate the helpful comments of the reviewer. Non-small-cell lung carcinoma (NSCLC) model cell lines A549 and H1299 have been widely used to study the molecular mechanisms of tumor initiation, progression and metastasis. (*Dev Cell.* 2015, 32: 318–334; *Cell.* 2016, 166: 1269-1281; *ACS Nano.* 2018, 12: 2332-2345). Previous studies also demonstrated Nestin regulates proliferation, migration, invasion and stemness of lung adenocarcinoma (*Int J Oncol.* 2014, 44: 1118-1130; *Chest.* 2011, 139: 862-869). High expression levels of Nestin are reported in lung cancer and Nestin expression is a prognostic indicator of a poorer survival probability in NSCLC patients (*PLoS One.* 2017, 12: e0173886). Thus, we used A549 and H1299 NSCLC cell line to investigate the relationship between nuclear Nestin and tumor senescence. In addition, we also detected Nestin expression in several other

cancer cell lines (including glioma, esophageal cancer and prostate cancer) and observed that Nestin also localized in the nuclei of these cancer cells, indicating the nuclear translocation of Nestin might be a universal phenomenon.

The corresponding data was present in Supplementary Figure 3e and 3f.

Minor points 2: The validity of the nestin in the nucleus should be confirmed with a 2nd antibody for IHC and knock down for immunofluorescence and subcellular fractionation (Figure 5g shows knock down combined with subcellular fractionation).

Response: Thank you for your kind suggestion. As requested, we performed IHC with another Nestin antibody (Millipore, ABD69). In consistent with our previous immunofluorescence and immunoblotting results, Nestin was also present in the nuclei of A549 and H1299 cell lines. **The corresponding data were added to Figure 3c and the previous Figure 3c was adjusted to Supplementary Figure 3c.** Also, we performed additional experiments and replace new figure for the Nestin and lamin A/C colocalization in Figure 5h (immunostaining with anti-lamin A/C and the new anti-Nestin antibody). The results further demonstrate Nestin knockdown triggered lamin A/C dispersion from the nuclear lamina.

Minor points 3: Figure 4A is lacking a key control – omit the flag-tagged nestin from the anti-flag IP.

Response: We apologize for the carelessness. As suggested, we performed immunoprecipitation experiments again and further confirm the interaction between Nestin and lamin A/C. **We added the corresponding data in Supplementary Figure 4a.**

Minor points 4: Figure 4b. Can the authors confirm the lamin A/C – nestin interaction by IP of endogenous nestin?

Response: We have already tried this approach. Unfortunately, three commercial Nestin antibodies for immunoprecipitation (santa cruz, sc-33677; LSBio, LS-B2851; abcam, ab6320) have been used to immunoprecipitate endogenous Nestin, but none of

them worked well. Therefore, we are unable to confirm the lamin A/C – Nestin interaction by IP of endogenous Nestin. Therefore, we verify the lamin A/C – Nestin interaction by IP of endogenous lamin A/C and exogenous Flag-Nestin co-IP.

Minor points 5: Figure 4d. Is the C terminus of the rod domain sufficient for interaction with nestin, or only necessary?

Response: We appreciate the helpful comments of the reviewer. To answer this question, we further generated a Myc-Nestin construct without the C terminus of the Nestin rod domain ($\Delta 182-313$), overexpressed it separately with the other Myc-Nestin construct (Full Length, FL & 182-313) and immunoprecipitated with anti-lamin A/C antibodies. In consistent with our results in Figure 4e, fragments contain the C terminus of the Nestin rod domain (FL & 182-313) retained the ability to bind lamin A/C. In contrast, fragment $\Delta 182-313$ did not have the ability to bind lamin A/C. Therefore, the C terminus of the rod domain is **both sufficient and necessary** for Nestin to interact with lamin A/C.

Attached/Reviewer Figure 2 Nestin interacts with lamin A/C.

(a) Schematics of the utilized Myc-Nestin constructs.

(b) HEK293T cells transfected with the indicated constructs were subject to

immunoprecipitation with anti-lamin A/C antibodies.

Reviewer #3 (Remarks to the Author):

In this manuscript, Zhang et al. set out to study subcellular locations and interactions with lamin A/C for Nestin. They have listed a few new findings: first, classical nuclear localization signals were found at the Nestin rod domain. Second, the rod domain can directly interact with lamin A/C. Third, the lack of Nestin causes the phosphorylation and nuclear export of lamin A/C.

There are some major and minor issues as follows that need to be addressed before a recommendation of publication can be made:

Major Point 1: In the introduction, the authors just simply listed previous findings, which makes the reading experience not very enjoyable. This reviewer suggests that it could be better organized by classifying these findings with a sound and concise flow.

Response: Thanks for your kind suggestions. As suggested, we reorganized the introduction. **First of all**, we briefly described the traditional understandings of Nestin as adult stem cell marker and diagnostic tumor marker. **Next**, we emphasized the recently reported multi-faceted roles of Nestin in cellular processes. **Finally**, we propose the novel function of Nestin in the nucleus.

Major Point 2: Circularity of nuclear shape is used as an important and sole indicator for quantitatively distinguishing the normal and abnormal nuclear shapes with or without knockdown of Nestin in different cell lines. Although a previous publication was referenced and a few words were provided in Methods, a more detailed introduction is still needed to clarify the data process. Also, if this is a well-accepted approach, this reviewer is curious why only one previous publication was cited; if not, a serious question is how reliable the approach is. If this question cannot be well addressed, the rest studies in the manuscript would start shaking.

Response: We are sorry for the unclear description about the data process for nuclear circularity. Specifically, A549 and H1299 cells were fixed in with 4% formaldehyde

and images of nuclei were acquired using a Zeiss LSM-800 laser scanning confocal microscope. Then, we used CellProfiler software to quantify nuclear circularity and nuclear area from H2B-GFP expressing or DAPI staining grayscale pictures. Shape features of nuclei were determined using the “identify primary objects” followed by the “measure object size shape” and “export to spread sheet” module. Among these features, we calculated the nuclear circularity or contour ratio as: $4\pi \times \text{area}/\text{perimeter}^2$ (The contour ratio has a maximum value of 1 for a circle and decreases as the nuclear shape becomes increasingly convoluted). **As suggested, we added the detail information in the "Methods" section.**

In addition, we apologize for citing only one previous publication of nuclear circularity. Actually, lots of studies used this indicator to quantify nuclear shape changes (*EMBO J.* 2012, 31: 1080-1094; *Science.* 2014, 344: 527-532; *Nat Commun.* 2017, 8: 16013; *J Cell Biol.* 2005, 170: 781-791). Besides nuclear circularity, we also calculated three typical examples of nuclear shape alterations and scored the percentage of nuclear blebbing as shown in Figure 1b (*Nat Commun.* 2017, 8: 16013; *Sci Transl Med.* 2011, 3: 89ra58). **As suggested, we cited these references in the revised manuscript.**

Major Point 3: Another serious concern is about the NLS-dependent nuclear-cytoplasmic localization of Nestin. There are several confusing aspects. First, the information of software and data process used to predict weak or strong NLSs is insufficient to know if a score of ~4.2 means weak or strong signals. Second, the authors named all signals (NLS1, NLS2, NLS3) as NLSs, but the tests on NLS1 and NLS3 indicated that these signals caused Nestin to stay primarily in cytoplasm. So, these signals might be NESs rather than NLSs? Finally, an accurate and standard approach is to use Cytoplasm/Nucleus intensity ratio of GFP-tagged proteins to quantitatively determine their subcellular locations. For example, it's hard to tell how big the difference is between F4 and F5. Also, there is no clue how a conclusion that Nestin only stays at NE for F2 test was drawn.

Response: We appreciate the helpful comments and advice. Generally, many

researchers used the score to predict the possibility of nuclear localization: higher NLS scores (>6) indicate **stronger NLS** activities, and candidate with a NLS score of 4-6 is considered to contain **moderate NLS** localizing to both the nucleus and the cytoplasm (*Inflamm Res.* 2016, 65: 895-904; *FASEB J.* 2014, 28: 3480-3493). Thus, the score of ~4.2 predicts that **Nestin might have a moderate NLS signals**. Just like the reviewer's opinion, the information of software and data process used to predict weak or strong NLSs is insufficient. It is still necessary to do gain or loss-function experiments to confirm whether these target sequences are putative NLSs. Because of this, we further chose three sequences, whose scores were more than 4, to investigate their subcellular localizations by confocal fluorescence microscopy. The results showed that GFP fused to fragment contains NLS2 yielded both nuclear and cytoplasmic signals, which was similar with our bioinformatic prediction.

To address the reviewer's second concern, we carried out another bioinformatic analysis (<http://www.cbs.dtu.dk/databases/NESbase>) to clarify whether these signals might be NESs. The results showed that Nestin sequence had no putative NESs. Thus, NLS1 and NLS3 might not be the NESs.

For the reviewer's third question, we analyzed the Cytoplasm/Nucleus intensity ratio (CN ratio) of GFP-tagged proteins to quantitatively determine their subcellular locations. The results showed that GFP fused to the F6 and F2 fragments had a CN ratio less than 1, indicating a prone distribution to the nuclei. GFP fused to the F1 and F4 fragments also yielded nuclear signals, though at lower levels (CN ratio was ranging from 1 to 2). Whereas, GFP fused to F3 and F5 fragments localized to the cytoplasm as evidenced by a CN ratio more than 20. With the quantitatively evaluation, we believe the subcellular locations between different fragments is much accurate now. **As suggested, we added the corresponding data in Figure 3f.**

Finally, we apologize for the unclear description about the localization of F2. We further analyzed its subcellular localization by Zeiss LSM-800 laser scanning confocal microscope and got 3D confocal fluorescent images. As shown below, actually, F2 is mainly deposited at the periphery of outer nuclear envelope. **Accordingly, we replaced the term about the localization of F2 with the more**

accurate description as “at the periphery of outer nuclear envelope (PE)” in Figure 3d and its related figure legend.

Attached/Reviewer Figure 3 Confocal images of GFP-Nestin fusion protein. 3D confocal images of F2 in A549 and H1299 cell lines. Nuclei were marked by DAPI (blue). Scale bars, 5 μ m.

Major Point 4: Finally, this reviewer thinks that the section of discussion could be better handled by concisely highlighting the major results and their impacts on the specific field. The current version is too sparse and unclear.

Response: Thanks for your kind suggestions. Accordingly, we revise the discussion section to make the points more clear.

Reviewer #1 (Remarks to the Author):

The authors have sufficiently responded to the reviewer's comments and suggestions, and the manuscript is in condition for acceptance.

Reviewer #2 (Remarks to the Author):

The authors have responded well to all my critiques. This MS is now acceptable for publication in my opinion.

Reviewer #3 (Remarks to the Author):

All my questions and concerns have been well addressed in the revised manuscript.